# A Review of the Vaporization Behavior of Some Metal Elements in the LPBF Process

**DOI:** 10.3390/mi15070846

**Published:** 2024-06-29

**Authors:** Guanglei Shi, Runze Zhang, Yachao Cao, Guang Yang

**Affiliations:** College of Mechanical, Hebei University of Science and Technology, Shijiazhuang 051432, China; shiguanglei@hebust.edu.cn (G.S.); zrz17754110607@163.com (R.Z.); yachao.cao@hebust.edu.cn (Y.C.)

**Keywords:** metal additive manufacturing, vaporization behavior, by-products, LPBF

## Abstract

Metal additive manufacturing technology has developed by leaps and bounds in recent years; selective laser melting technology is a major form in metal additive manufacturing, and its application scenarios are numerous. For example, it is involved in many fields including aerospace field, automotive, mechanical processing, and the nuclear industry. At the same time, it also indirectly provides more raw materials for all walks of life in our country. However, during the selective laser melting process, due to the action of high-energy-density lasers, the temperature of most metal powders can reach above the vaporization temperature. Light metals with relatively low vaporization temperatures such as magnesium and zinc have more significant vaporization and other behaviors. At the same time, during the metal vaporization process, a variety of by-products are generated, which seriously affect the forming quality and mechanical properties of the workpiece, resulting in the workpiece quality possibly not reaching the expected target. This paper mainly interprets the metal vaporization behavior in the LPBF process and summarizes the international research progress and suppression methods for vaporization.

## 1. Introduction

As far as the current world development level of the manufacturing industry is concerned, the manufacturing processes of metals are generally divided into three categories, namely additive manufacturing, subtractive manufacturing, and equal-material manufacturing, as shown in Figure 1 [1]. Additive manufacturing technology was first developed in the 1980s and, before that, it was also called rapid prototyping manufacturing [2]. Metal additive manufacturing, also known as metal 3D printing, has great advantages compared to traditional subtractive manufacturing techniques in the manufacture of complex integral components, lightweight engineering and material utilization [3,4]. It no longer depends on the cutting tools, fixtures, molds and various cumbersome processing procedures required by traditional machining and can quickly and precisely manufacture parts of any complex shape, which not only shortens the processing cycle but also greatly saves raw materials [5,6,7,8,9]. At the same time, the molten pool generated by selective laser melting technology in additive manufacturing technology is significantly smaller than the casting pool and welding pool [10]. Therefore, additive manufacturing technology plays an important role in many fields such as aerospace [11,12], medicine [13,14], the automotive field [15], mechanical processing, and the nuclear industry. Therefore, many scholars regard it as the representative technology of the “third industrial revolution” [16].

Among metal additive manufacturing technologies, the two most widely used forms are direct energy deposition (DED) [17,18] (Figure 2 shows the common powder feeding method of DED) and powder bed fusion (PBF) [19], and in PBF, it is further divided into selective laser melting (LPBF) with a continuous laser as the energy source (accounting for 82% of the PBF market in 2016) [20] and electron beam powder bed melting (EBPBF) with an electron beam as the energy source. The equipment and structural composition of LPBF are shown in Figure 3 [21]. In terms of applicable materials, nickel (Ni)-based, titanium (Ti)-based, aluminum (Al)–copper (Cu)-based, iron (Fe), zinc (Zn), tantalum (Ta)-based, molybdenum (Mo)-based alloys and alloys such as 316L and 304 stainless steel all show good results in the LPBF process, and the formed parts are widely used in various fields [22,23,24,25,26,27,28,29,30,31,32,33].

Although its range of raw materials is wide, in the LPBF process, the laser acts directly on the powder particles, so the condition of the molten pool directly determines the surface quality and physical properties of the formed part. Therefore, it can be considered that the comprehensive effect of the multi-physics field in the LPBF process is actually the result of the combined effect of interfacial heat transfer and temperature [34,35,36]. Therefore, for almost all alloys, during the forming process, affected by the dynamic of the molten pool, many phenomena such as spheroidization, keyhole, defects, and spattering are still inevitable [37].

For some light metals with relatively low vaporization temperatures such as magnesium, zinc, and manganese, the vaporization phenomenon is extremely significant under a high-energy laser density, and a large amount of smoke and dust appear in the forming chamber, as shown in Figure 4 [38]. The generated plume blocks the laser, leading to fluctuations in the molten pool, which directly cause many defects such as pores, bubbles, microcracks, residual stress, and deformation in the formed parts [39,40,41,42]. At the same time, the generation of vapor recoil pressure forms a cavity, called a keyhole [43], and after the cavity is closed, it may also form defects such as pores and bubbles (Figure 5), which then cause phenomena such as splashing, keyhole, and erosion, resulting in molten droplets or large-particle powders in the molten pool splashing to different positions on the powder bed, thereby affecting the stability of the molten pool [44,45]. Precisely because of this, studying the formation mechanism of the vaporization by-products of metal powder in the LPBF process and the various negative impacts it causes has already become the top priority in the current development process of LPBF technology. The common means to solve these problems is usually by optimizing process parameters such as the laser power, scanning speed, scanning radius, spot diameter, layer thickness, dust removal wind field, etc. [46,47,48,49,50,51,52,53,54].

According to the current research situation, due to the lack of in-depth understanding of the vaporization process, it is an extremely challenging task to completely suppress the generation of by-products. Pure experimental methods such as molding experiments and visual measurements can only be used to conduct qualitative research on the LPBF vaporization phenomenon and analyze the impact of vaporization products on the quality of formed parts but cannot be used to conduct quantitative research on information such as the quality, flow rate, and generation rate of vaporized by-products. In order to deeply understand the physical and motion processes behind it, in recent years, more and more scholars have devoted themselves to conducting research on this aspect. By establishing a mesoscopic platform through simulation and modeling, it is more intuitive to explore the interaction between the laser and the powder, analyze the metal vaporization process, and promote the innovation and development of LPBF technology. However, most numerical simulation models cannot perfectly replicate the interaction process between the laser and the powder, and they always ignore problems such as plume generation and multi-physics field coupling. Through summarizing the research results of various scholars, this article comprehensively discusses the generation mechanism and suppression methods of various by-products in the vaporization process and further promotes progress in solving the vaporization problem.

The logical structure of this article is as follows: in Section 2, the mechanism of metal vaporization in the LPBF process is introduced and some by-products in the vaporization process, such as spattering, denudation, keyhole, etc., are summarized; in Section 3, the research results of the vaporization phenomenon in the LPBF process in the current international research are summarized from the experimental means and numerical simulation means, respectively, and the advantages and disadvantages of each method are generalized. In Section 4, the methods to suppress the vaporization behavior are summarized, respectively, from the aspects of the laser energy density, atmospheric environment, and circulating air flow; in Section 5, the conclusion is provided.

## 2. Metal Vaporization in LPBF Process

### 2.1. Metal Vaporization Mechanism

In the LPBF process, some scholars believe that the vaporization phenomenon is a situation that can be avoided and should be removed as much as possible through certain means [56,57], while other scholars believe that the vaporization behavior is an inherent part of the LPBF process and should be carefully adjusted [55]. The metal vaporization phenomenon induced by the laser is an important physical phenomenon in the molding process, which affects the stability of the molten pool and the molding quality of the workpiece to a certain extent [58]. And because the laser action time is very short during the working process and the energy density is very large (reaching approximately 105–107 W·cm^−2^ [59]), the cooling rate is very high (approximately 105–106 K/s) and the latent heat from the vaporization of metals is generally approximately 10 times that of the latent heat of melting; thus, for some light metals with a lower boiling point, the exposed powder particles are extremely prone to melting and evaporation [60,61,62]. When the powder temperature is heated to the vaporization temperature, some metal atoms “escape” from the surface of the powder by overcoming the bond energy between the atoms. When the incident laser further irradiates the “escaped” metal vapor, it ionizes itself and then forms a plasma [63]. The plasma, vapor, and atmosphere form a plume, and this part of the metal particles is called metal vaporization (and also evaporation, boiling, or volatility). In short, the significance of vaporization lies in the phase transition process of metal powder from the solid and liquid phases to the gas phase [38].

### 2.2. LPBF Metal Vaporization By-Products

In the LPBF process, vaporization phenomena occur in almost all metal materials, but the metal vaporization phenomenon itself does not cause excessive damage to the molten pool state or workpiece quality. What really damages the molten pool state and workpiece quality are various by-products in the metal vaporization behavior.

After the metal vaporization phenomenon occurs, the resulting back pressure forms a cavity on the molten pool, which we call a keyhole (as shown in Figure 6 [64]), and the plume formed by vaporization causes some particles to be involved in it, resulting in the denudation phenomenon; the involved particles eventually fall within the powder bed and melt the track, thereby causing the spattering phenomenon [65,66,67].

Many scholars have revealed the formation mechanism of the keyhole through different models; that is, the formation of the keyhole is closely related to the recoil pressure [68,69,70,71]. It is worth mentioning that the keyhole itself is an important and inevitable physical process in the LPBF process, and what really affects the workpiece quality is the pores formed after the keyhole is closed [72]. The formation of pores inevitably affects the internal quality of the final workpiece and reduces its mechanical properties. In response to the pores, scholars have introduced a definition, that is, “normalized enthalpy”, and they have set a threshold for this process. If it is lower than this threshold, it will cause insufficient melting of the molten pool, and if it is higher than this threshold, it will lead to the formation of more pores [68,72]. In addition, other scholars have also conducted relevant research on the keyhole phenomenon caused by metal vaporization behavior in the LPBF process [73,74,75,76,77,78,79].

Spattering (as shown in Figure 7 [80]) and plume (as shown in Figure 8) are two other by-products in the LPBF metal vaporization process [81]. The formation of spattering and plume is an extremely complex process. The generation of spattering causes large-particle powders and molten droplets to fall within the melt track, resulting in an increase in surface roughness and even the introduction of defects. At the same time, it also causes the temperature of the molten pool to be uneven, affecting the stability of the molten pool. The plume affects the actual power of the laser acting on the powder bed [82,83], and both spattering and plume may cause the laser to be contaminated, seriously affecting the accuracy of the laser. Therefore, these two by-products have a direct adverse impact on the quality of the formed parts. Spattering is usually divided into two types [84] and is often regarded as a significant unstable factor affecting the molten pool [85]. The first type is considered to be the molten droplets flying out from the molten pool during the forming process, which is mainly driven by the vapor recoil pressure [86], and the second type is powder particle spattering, which is mainly generated by the denudation effect and is finally involved in the vapor [87]. Therefore, in order to avoid the impact of spattering on the molten pool, some scholars are committed to developing spattering-free models, but they are only applicable to individual metal materials with relatively large limitations [88,89]. The plume refers to evaporated smoke in the metal LPBF process, and its composition is relatively complex.

The denudation phenomenon in the LPBF process (as shown in Figure 9 [90]) refers to the redistribution of powder due to some reason, and some particles enter the molten pool to form a “denudation area” [91,92]. The denudation caused by the metal vaporization phenomenon in the LPBF process destroys the stacking state of the powder, thereby destroying the continuity and leading to defects in the molten pool. The formation of a zone without powder damages the adhesion between the part layers and leads to dimensional distortion. At the same time, the denudation phenomenon also causes some powders to undergo chemical reactions and be oxidized.

### 2.3. Section Summary

In the LPBF process, the metal vaporization phenomenon is an inevitable outcome. Its essence is a process in which metal materials undergo a transformation from the solid phase and liquid phase to the gas phase after being heated. In the metal vaporization process, some by-products such as plumes and spattering are generated, and some physical phenomena such as pores caused by keyholes and denudation also occur. These by-products and physical phenomena result in low workpiece quality and poor performance. We do not want such changes to happen, so we are always trying to find ways to curb their occurrence.

## 3. LPBF Metal Vaporization Phenomenon Research Progress

In the LPBF process, some scholars observe the generation process of by-products such as plumes, spattering, and keyholes by establishing mesoscopic platforms such as high-speed camera imaging [93,94], X-ray imaging [87,95], and high-speed schlieren imaging [80]. However, due to the difficulty in quantitatively measuring the metal vaporization products during the experiment, some scholars also adopt numerical simulation methods to quantitatively study the generation amounts of by-products and the like. In the existing research work, in addition to analyzing the influence of input laser energy and circulating shielding gas on the number and behavior of spattering [96,97,98,99], research on the generation behavior of keyholes under different energy densities has also been carried out [55]. However, due to technical limitations in many aspects such as software and hardware, the research results of most scholars more or less have certain imperfections.

### 3.1. Experimental Study on Vaporization Products

In recent years, as the monitoring technology has gradually matured, Marko Bärtl [100] and others have studied the relationship between surface tension and metal vaporization through high-speed cameras and electromagnetic levitation and other equipment. They claim that the metal with a smaller surface tension has a relatively smaller degree of metal vaporization, and it has been proved that the degree of metal vaporization of Mg is greater than that of Mn. At the same time, the phenomena of spattering and elemental burnout exhibited by Mg–Al alloys are more significant. This scholar and his team are also the first team to study the metal vaporization behavior from the perspective of surface tension. Marko Bärtl and others used electromagnetic levitation equipment to measure surface tension and used high-speed cameras to observe by-products such as spattering during the metal vaporization process, opening up a new direction. In the future, we may be able to regulate the surface tension of metal molten droplets by adding certain second-phase materials or other methods to metal materials, thereby avoiding the impact of metal vaporization on the quality of workpieces.

P.Bidare [80] and others built an in situ measurement platform for the LPBF process and used a high-speed camera to record the process of plumes and spattering in the LPBF process. As shown in Figure 10, it is observed through schlieren that the plume formed at the moment of laser incidence has a clear contour, and then a large number of particles rush out of the plume to form spattering. They claim that Fe vapor, plasma, and the induced argon gas flow determine the mode of powder denudation from near the molten pool, and the direction of spattering changes with the change in process parameters, and the speed of the generated plume is proportional to the laser power. The experimental platform built by P. Bidare and others is comprehensive. It not only has a high-speed camera to record the spattering process, but also makes it possible to visualize the plume through schlieren imaging, and through the combination of experiment and numerical simulation, it elaborates, in detail, the molten pool dynamics and the kinematics and dynamics of products such as plumes and spattering in the LPBF process, making great contributions to the research on LPBF metal vaporization.

Zhirnov [101] and his colleagues successfully captured the process of steam eruption through high-speed CCD camera shooting. By analyzing the captured images, it was found that the steam plume is approximately 4–5 mm long and approximately 1.5 mm wide, and it was proved that the speeds of steam, droplets and powder particles are related to the calculated flow field. Zhirnov and others also used high-speed cameras as a means to monitor, in real time, the morphology and movement process of the plume in the LPBF process under different time periods and different process parameters, laying the foundation for subsequent regulation of the gas phase flow to suppress metal vaporization.

Hoppe [102] and Jauer [103] concluded through observing schlieren imaging that the plume ejected from the molten pool accumulates above the molten pool and interacts with the laser. Once a certain critical level is reached, the molten pool becomes unstable, which further proves that when the plume concentration in the forming chamber reaches a certain critical value, the fluctuating laser will cause the molten pool to become turbulent.

The research of several scholars has shown that the plume is a major product in the LPBF metal vaporization process, and its influence on the molten pool cannot be ignored. We can try to minimize its impact on the quality of the formed parts by changing the scanning rate and the amount of dust removal. However, most of the models built by scholars do not include any interaction between melt flow and the atmosphere [104].

Zhang [105] and others used a high-speed camera with a visible band cut-off filter to monitor the plume and spattering signals, and they extracted their characteristics from multiple aspects and proposed the SVM classification method. As shown in Figure 11, the ability to identify the quality level is evaluated by using the characteristics of each object and three different objects. Through the extraction of the signals, they found that the thermal radiation and plasma radiation generated by the ejected metal vapor generates a “plume”, and the plume vapor is ejected from the inside of the keyhole, so the radial ejection shape of the plume is related to the keyhole wall and the exit contour. Zhang and others used the SVM classification method to divide the plume, spattering and other signals into different levels to study the degree of metal vaporization under different signals more deeply. This method has high accuracy and a strong generalization ability and is not sensitive to outliers, so it has strong adaptability in the LPBF process and expands the monitoring means of by-products.

V. Gunenthiram [96] and others conducted LPBF experiments using 316 L powder and recorded videos at an angle of 60°; they found that the gas flow caused by vaporization is one of the main reasons for spattering, and it was proved that the amount of spattering also increases as the energy density increases. You Wang [106] and his colleagues conducted a series of studies on Haynes 230 through in situ monitoring methods. They noted that the states of spattering and plume in the LPBF process change with the temperature. When the input energy is low, the molten pool does not form a depression, and the plume and spattering are generated almost perpendicular to the surface, while after the depression is formed, the angle between the plume and spattering and the surface will be greater than 90°. Several scholars have studied the impact on the by-products in the LPBF process from the perspective of incident energy density, making it a reality to suppress metal vaporization by regulating the incident laser process parameters.

Di Wang [99] and others used a high-speed camera to monitor the dynamic changes in spattering under different energy densities. They claimed that the stronger the laser energy density, the greater the amount of spattering and the greater the impact on the quality of the liquid phase of the formed parts. According to the SEM images of the spattering particles, they can also be classified into three categories: As shown in Figure 12b, Type I spattering is spherical spattering with a smooth surface; as shown in Figure 12c, Type II spattering is spherical spattering with a rough surface; as shown in Figure 12d, Type III spattering is an irregularly shaped granular powder on the surface. In this research, Di Wang and others classified the types of spattering specifically, allowing readers to understand the morphological changes in spattering after it flew out of the molten pool surface, providing a basis for further research on the dynamics of spattering in the future.

In addition to this, other scholars have also made great contributions to the research on spattering in the LPBF process [107,108,109,110,111,112,113,114]. Although many scholars have conducted in-depth research on the spattering phenomenon, due to the small volume of the spattering objects and the high speed of spattering, there are still certain challenges in the research on spattering so far [115,116].

Martin et al. [117] elucidated the formation mechanism of the keyhole during the laser turning period through in situ X-ray technology. They claimed that when the laser moved away, the keyhole collapsed, and the molten metal and argon filled the voids. During the rapid cooling process of the molten pool, the remaining argon forms new voids, which then affect the quality of the formed parts. At the same time, they proved that by adjusting the laser parameters, powder properties, and processing conditions, the pore formation caused by the keyhole could be reduced. R. Cunningham [55] et al. restored the formation process of the keyhole through X-ray technology (as shown in Figure 13). They pointed out that there was a large amount of liquid metal and vapor distributed in the depression and revealed some mechanisms related to the shape of the keyhole.

Professor C. Zhao [118,119] used high-speed synchrotron hard X-ray imaging and diffraction techniques to conduct real-time monitoring of the laser powder bed fusion (LPBF) process of Ti-6Al-4V. Through experiments, it was found that the high-energy density laser caused rapid evaporation of the powder, and the ejection behavior of powder particles was observed. By observing different sections of the molten pool, the formation process of the keyhole was clearly captured, and the boundary definition of the keyhole porosity was clarified. It was found that the keyhole would only form when the depth of the keyhole was greater than a certain threshold. This was the first time that the time required for keyhole closure had been experimentally measured, making a great contribution to the research progress on keyholes in the LPBF process. In [120], the author used X-ray technology and, based on previous scholars’ research, also discovered three formation mechanisms of the keyhole pores (as shown in Figure 14): (1) pores captured by surface fluctuations, (2) pores formed due to fluctuations in the depression zone when the depression zone was relatively shallow, and (3) pores formed by cracks. Several scholars have explained the formation and transfer dynamics problems of the keyhole and pores through X-ray experimental methods and have also summarized the differences between and generation conditions of different types of keyholes, laying a huge foundation for subsequent research to eliminate the negative impacts brought by the keyhole.

Hui Chen [121] et al. measured powder denudation through SEM means and concluded that the particles near the metal evaporation center flowed to the melt track at a significantly faster speed and the drag force of the ejected metal vapor on the particles was much greater than the resistance generated by the inward ambient airflow. This indicates that the denudation zone is mainly generated near the molten pool; the farther away from the molten pool, the smaller the negative impact brought by the denudation. Matthews [90] and others took high-speed imaging as a means. Through the research on the powder around the scanning path, it was concluded that the powder denudation near the melt track was caused by both the vapor flux generated by evaporation and the inward shear flow of the shielding gas. The process of powder denudation has also been interpreted through high-speed imaging and optical microscopy. As shown in Figure 15, at 200 microseconds, some particles (blue trajectories) collided with the molten pool and were thus drawn into the molten gold pool. Before this, there was not a clear picture of denudation in the international community. The author and his team examined clear denudation phenomena for the first time, proving the influence of the circulating airflow on the molten pool and denudation, making a great contribution to the research on denudation phenomena.

Yang Du [122] et al. and Makowska M [123] and his team both found, through experimental exploration of the relationship between laser energy and the width of the molten pool, that not only the width of dissolution but also the degree of denudation increase as the laser energy increases. Excessively high energy density leads to excessive denudation. L. Kaserer [124] et al. found that there is also a relationship between the denudation effect and the atmospheric pressure. Between 1 mbar and 10 mbar, the width of the denudation zone decreases as the pressure increases. Through descriptions by several scholars, we realize that the denudation phenomenon, like other vaporization by-products, also has dynamic changes, and it also changes with factors such as energy density, atmospheric pressure, and airflow size.

### 3.2. Simulation of Vaporization Products

The above scholars who study vaporization problems through experimental methods mainly conduct qualitative analysis on vaporization phenomena and cannot quantitatively measure the specific loss of elements in the experimental process. To solve this problem, more and more scholars adopt the method of numerical simulation to devote themselves to a more in-depth and thorough study of the vaporization behavior of metal powder materials in the LPBF process.

Yi luo [125] studied magnesium alloy by establishing an evaporation model and obtained that the vaporization time of Mg, Al, and Zn was Mg < Zn < Al from the direction of the vaporization time, and Mg vaporized prior to Zn. This scholar and his team identified the problems of the vaporization time and vaporization degree of metal powder materials through simulation. According to the simulation results, no matter the vaporization degree or the sequence of vaporization, the metal element with the lower vaporization temperature is prioritized. However, thermal physical parameters such as the density, thermal conductivity, and specific heat of the magnesium alloy material in the established model were all set as constants, which was contrary to common sense to some extent, and this behavior might have a certain impact on the accuracy of the simulation results.

Wei Kaiwen and J Jakumeit [126] et al. used the method of numerical simulation to integrate the relationship formulas related to element burning loss and metal surface evaporation flux, laying a certain foundation for subsequent scholars to conduct quantitative analysis. However, using the numerical simulation method will inevitably not fully consider the actual situation, so there is a certain error. Taking the model built by J Jakumeit et al. as an example, in this model, they regarded metal vapor as an ideal gas and alloy materials as a single pure metal material. At the same time, the absorption, reflection and shielding of the laser as well as the thermal radiation and emissivity of the materials are difficult to measure. However, it is worth mentioning that the volumetric heat source method was used to further diffuse the heat into the powder layer interior, avoiding the phenomenon that the laser energy is only deposited on the powder layer surface.

Alexis [127] and others studied the influence of vaporization phenomena on the dynamics of the molten pool through the numerical simulation method (Figure 16). Their research results indicated that when the powder temperature exceeded the vaporization temperature, the recoil pressure formed would force the molten pool shape to deflect. Moreover, as the laser energy increased, the formed keyhole tilted, thus resulting in the laser being concentrated inside the keyhole and causing the gas to be trapped inside the keyhole. However, it needs to be noted that in the model they constructed, the powder layer was assumed to be a continuous area, which might have a certain influence on the accuracy of the simulation results, because in the actual situation, the powder layer could not be entirely continuous.

Most scholars often neglect the impact of vaporization effects on the molten pool during numerical simulations. However, Yu-Che Wu [128] et al. used a Gaussian surface heat source to numerically simulate the LPBF process and treated the powder bed as a discrete body. They found that the formed melt track was wide but shallow without considering vaporization, while the melt track formed considering vaporization was narrow but deep, and the vaporization effect reduced the surface temperature. Parivendhan et al. [129] used numerical simulation to study the molten pool dynamics by modeling the powder bed and the thermal fluid, but a drawback of their approach was the insufficient calculation of the plume during the evaporation process.

Hui Chen [65] and others established a multiphase flow numerical model through two-way coupling of DEM and FVM to explore the spattering and denudation phenomena in the LPBF process, as shown in Figure 17. They claimed that the initial velocity of the metal vaporization jet was very fast, but the velocity decreased rapidly after the jet, and the jetted vapor was the main cause of spattering and denudation, and they also explained the formation process of spattering and denudation. However, due to technical cost reasons, the influence of the liquid phase and phase change on it was not considered in this simulation.

Yunfu Tian [130] and others found, through numerical simulation, that when the input laser energy is insufficient, it will lead to keyhole turbulence and the formation of pores, while proper energy input will alleviate this phenomenon. Mohamad Baya [131] and others used the commercial software Flow3d to replicate the formation process of the keyhole of the Ti6Al4V alloy material in the LPBF process. They claimed that in the process of the keyhole extending downward, the molten liquid flows downward and the laser cannot penetrate so far, which leads to a large amount of energy loss and also causes a local temperature reduction. Therefore, the local increase in surface tension and the significant reduction in recoil pressure eventually lead to the formation of the keyhole, and when the temperature drops, the merging of multiple pores will lead to the formation of a new pore (as shown in Figure 18). However, the model they built cannot achieve ray tracing and cannot well simulate the repeated collision process of the laser in the keyhole. Therefore, the formed keyhole will theoretically have a certain error compared to the actual situation.

Matthews [90] and his colleagues found, through finite element simulation, that the ambient pressure also affects the degree of denudation to a certain extent. Between one atmospheric pressure and an argon pressure of 10 Torr, the denudation width is negatively correlated with the ambient pressure. However, the simulation model they built ignores some physical fields, which may lead to a certain error between the simulation result and the actual result. But we can still conclude that by adjusting the magnitude of the ambient pressure, the negative impact caused by the denudation effect can be reduced.

Besides the scholars and their teams mentioned above, there are also some scholars who have conducted in-depth studies in the direction of the metal vaporization behavior in additive manufacturing [132,133].

### 3.3. Section Summary

These scholars all explore and study the direct characterization of the metal vaporization behavior in the LPBF process through different experimental methods. The most direct method is to establish a mesoscopic platform by using means such as high-speed cameras, X-rays, and schlieren imaging. The high-speed camera can accurately capture the changes in the LPBF process, but it can only observe the surface and cannot observe deep inside. Schlieren imaging is a non-contact surface detection method that will not cause damage to the workpiece and is conducive to maintaining the integrity of the workpiece. It is also simple to operate and can help us capture information such as the contour of the plume. X-rays make up for the deficiencies of high-speed cameras and schlieren imaging. It can penetrate the powder surface and then observe the internal imaging, and it can discover defects efficiently and is helpful for a comprehensive assessment of work quality. However, the penetrating power of X-rays for different materials is also different, and there is a certain amount of radiation.

Although numerical simulation can restore microscopic phenomena to a certain extent by means of mesoscopic methods, considering the time cost and technical limitations, there are still some points that need to be ignored in the models built by various scholars. For example, in some models, the powder is regarded as a continuum, while in others, the powder is regarded as a discrete model; and in some models, only spattering is considered, while in others, only denudation is considered. Even to date, there has not yet appeared a numerical simulation model that couples the plume movement with the melt pool dynamics, which also means that the numerical simulation cannot completely, actually, and systematically restore the actual process. However, as more and more scholars adopt the method of numerical simulation to explore the vaporization phenomenon and its influence in the LPBF process, the numerical model is also being rapidly improved, promoting the rapid development of LPBF technology.

## 4. LPBF Process to Suppress Vaporization

At present, the suppression of the negative impacts caused by the vaporization phenomenon mainly starts from two aspects. One is to find the laser parameters with the lowest evaporation degree by adjusting the laser energy density, and the other is to adjust the processing atmosphere, including optimizing the atmosphere passage, adjusting the magnitude of the atmosphere flow rate, and selecting different inert gases for different materials. Of course, in addition to these, there are also some niche methods, such as changing the particle size of the powder and adjusting the properties of the powder.

In Section 3, it is stated that different energy densities lead to changes in the degree of spattering, the size of the plume gas mass, the offset position, the degree of denudation, etc. Therefore, by adjusting the laser power and scanning speed, the negative impacts brought by the vaporization phenomenon can be suppressed [38]. The formula for the energy density of the laser is as follows:(1)$$E=\frac{P}{v\times h\times t}$$
where *E* is the energy density (J-m^−3^), *P* is the laser power (W), v is the scanning speed (m-s^−1^), h is the scanning distance (m), and t is the layer thickness (m).

In short, the suppression of the vaporization phenomenon can generally be started from the following aspects:(1)Adjust the laser energy density;(2)Adjust the flow speed and flow mode of the circulating air flow;(3)Adjust the size of the powder particle size;(4)Adjust the type of atmospheric environment and atmospheric pressure;(5)Adjust the composition of the alloy powder and change its properties.

### 4.1. Suppression of Vaporization by Energy Density

Yin Bangzhao [134] and others stated that a high energy density leads to intensified metal vaporization of Mg powder inside the molten pool, and, as shown in Figure 19d, a large laser energy density also causes an increase in internal stress and deformation, thereby deteriorating the internal molding quality of the molded part. Long Yuhang and others [135], through numerical simulation of the LPBF process of aluminum alloy, found that the metal vaporization of Al caused by a lower scanning speed drives the temperature diffusion of the molten pool, which increases the fluctuation and instability of the molten pool surface.

Yu Qin [136] and others studied the vaporization behavior of the Zn element and modeled it. By evaluating the relationship between the input energy and the vaporization rate, it was concluded that as the energy density increases, the amount of vaporization increases, and at the same time, the penetration power of the laser also increases, the formed plume significantly increases, and the quality loss increases. Some scholars have also found that the scanning speed has a greater impact on metal vaporization compared to the laser power [137]. Zhao [119] and others have also found that the porosity and the maximum pore diameter caused by the keyhole are both related to the scanning speed.

To sum up, as demonstrated by research by multiple scholars, lasers with different energy densities indeed have an impact on the degree of metal vaporization in the LPBF process. Usually, the greater the energy density, the more heat that will be input into the molten pool, which also leads to higher temperatures of the metal powder and more intensified metal vaporization. Therefore, for different materials, it is necessary to find appropriate process parameters to ensure that the energy density takes a moderate value, that is, neither too large nor too small. In this way, the negative impacts brought by the metal vaporization phenomenon can be suppressed to a great extent.

### 4.2. Suppression of Vaporization by the Atmospheric Environment

Besides energy density, the atmospheric environment is also one of the significant factors influencing the vaporization phenomenon. Introducing inert gases and circulating airflows is an effective measure to eliminate the by-products caused by vaporization in the LPBF process [138]. Figure 20 [136] indicates the working principle of the circulating airflow in the LPBF process. Generally, the wind speed of the circulating airflow acting on the surface of powder particles is between 2 and 5 m/s. As shown in Figure 21, a too-small circulating airflow cannot efficiently blow out by-products such as spattering and plume generated by vaporization out of the molding chamber or even out of the powder layer surface. However, a too-large wind speed will move the metal powder. Therefore, an appropriate circulating airflow is crucial for removing the by-products caused by the vaporization behavior.

Qin yu [136] and others established a numerical model to simulate the movement process of the circulating airflow. As shown in Figure 22a, when the dust removal air volume is insufficient, steady-state turbulence is formed above the molding chamber by the plume, which seriously blocks the laser. Figure 22b shows that sufficient dust removal air volume can blow the plume out of the molding chamber. From this, we can conclude that if the airflow intensity is too small, it cannot effectively remove the negative impacts caused by by-products such as spattering and plume. However, too large a circulating airflow will cause the distortion of the molding state. Therefore, an appropriate airflow intensity can effectively suppress the negative impacts caused by spattering and plume.

C. Pauzon [139] and his colleagues added helium to the high-purity argon shielding gas in the LPBF process of Ti-6Al-4V materials. The results, as shown in Figure 23, indicate that at least 60% of the spattering is reduced under pure helium, and the spattering decreased by 30% in the argon–helium mixture. S. Traore also studied the helium atmosphere. Through his research, it was found that the spattering phenomenon is more obvious under the argon environment, but the ablation phenomenon is more severe under the helium environment, resulting in greater energy loss (Figure 24) [140]. Therefore, the selection of the atmosphere is also a crucial link in the molding process, and an appropriate atmospheric condition can suppress the generation of vaporization by-products from the source and thereby improve the molding quality.

To sum up, the atmospheric environment and the circulating airflow are important links affecting the metal vaporization phenomenon in the LPBF process. Different atmospheric conditions have different effects under the same process parameters. However, at present, domestic and foreign scholars have produced relatively few reports on this aspect, especially on the aspect of optimizing the dust removal duct.

### 4.3. Other Methods to Suppress Vaporization

There are also some methods that will have certain effects on suppressing the vaporization behavior in the LPBF process, such as the particle size of the metal powder, the ambient pressure, the nature of the powder, and so on.

Chang Zhipeng [141], through their research on the generation of soot in the LPBF process of AZ91D magnesium alloy, found that by eliminating small particle size particles (below 20 microns) and larger particles (above 60 microns), the soot generated in the experiment was significantly reduced. This may be because the powder particles with smaller particle sizes usually have a larger specific surface area, leading to faster heat conduction, making the materials around the particles heat up more quickly and prone to local vaporization or volatilization, affecting the melting process of the materials under the laser beam. Larger particles may cause local overheating under the irradiation of the laser beam, because they can absorb the laser energy more effectively, resulting in changes in the vaporization of the materials and causing uneven surface morphology of the materials and affecting the forming quality.

Some other scholars start from the aspect of environmental pressure [142], changing it to increase the atmospheric pressure to achieve the suppression of the negative impact of vaporization in the LPBF process. This may be because the high-pressure environment will increase the vaporization temperature of the metal elements and thus reduce the vaporization phenomenon, while the low-pressure environment will also reduce the vaporization temperature of the metals and lead to an aggravated vaporization phenomenon. L. Deillon [143] and his colleagues found that adding 0.4 wt% of Ca element to Al–Mg–Sc alloy can change the properties of the alloy and effectively suppress the vaporization of the Mg element. The occurrence of this phenomenon may be because the added material has high stability or a high melting temperature, thereby reducing the vaporization rate of the metal powder and suppressing the vaporization tendency, or it may be that the added material will absorb the gas generated by vaporization to form new compounds and reduce the vaporization phenomenon.

### 4.4. Section Summary

In the LPBF process, suppressing the negative impacts caused by vaporization is a complex process, and certain effects can be achieved through a variety of different means. For different metal materials, there may be suitable respective suppression methods. However, for some lighter metals that are more prone to vaporization such as magnesium and aluminum, there have not been more reports on whether there are more effective suppression methods for them. In other words, for the currently known suppression means, they are almost universal for all metal materials. Therefore, to date, an optimal suppression scheme has still not been found, indicating that the research in this direction is a complex and long process.

## 5. Conclusions

LPBF is a complex technology involving the combination of multiple physical fields. The negative impact caused by the vaporization behavior of some elements in the process has an important constraining effect on the workpiece quality. In this article, we have discussed the vaporization phenomenon and its negative impacts in the LPBF process, such as problems like spattering, ablation, plume, and keyhole. By analyzing different methods to suppress the vaporization phenomenon, we found that these methods can, to a certain extent, reduce the negative effects brought by the vaporization phenomenon and play a positive role in the stability and forming quality of the LPBF process. However, although we have made certain progress, the research on LPBF vaporization behavior still faces many challenges. This article makes the following summary for the influence research and suppression means of metal vaporization behavior in the LPBF process:(1)At present, the commonly used experimental means to monitor the LPBF vaporization phenomenon include high-speed cameras, schlieren imaging, and in situ X-rays, each with its own advantages and disadvantages. However, these monitoring means can all help us further evaluate and improve the process parameters of the experiment.(2)For the existing numerical simulation models, there has not yet emerged a model that contains all vaporization products, such as being able to monitor both plume and spattering while also observing the ablation phenomenon caused by the airflow. And due to the existence of differences in spatial scales, there is also no numerical simulation model that can couple the motion state of the plume and the dynamics of the molten pool. Therefore, it is important to establish a perfect numerical simulation model.(3)Among the metal vaporization phenomena in the LPBF process, the occlusion phenomenon of the plume to the laser is a new research direction and also a weak direction in the current additive manufacturing field. For some easily vaporized light metal elements such as Mg, Zn, Mn, Al, etc., the plume generated in the LPBF process often forms a stable “cloud” on the forming chamber. This “cloud” absorbs a large amount of laser energy (up to 40%) to a large extent, seriously affecting the stability of the laser energy acting on the powder bed and also seriously affecting the quality of the workpiece. If the real-time occlusion effect of the plume on the laser can be predicted, and if the real-time regulation of the laser power can be achieved in the future, the combination of the two will greatly improve the quality and mechanical properties of the formed parts and will greatly promote the development of LPBF.(4)Metal vaporization is a necessary process in the LPBF process, and we cannot “eradicate” it from the root. In the future, we should further explore the methods and means to suppress vaporization, especially for magnesium and aluminum, which are more easily vaporized light metals, and their vaporization behavior often has a greater impact on the workpiece.

In conclusion, at present, the vaporization problem is still a problem that the metal additive manufacturing industry cannot effectively solve and is also a challenge we still face; especially for easily vaporized light metals, more technological breakthroughs should be sought in this regard. In the future, the metal additive manufacturing industry should continue to optimize the vaporization phenomenon in the LPBF process and its impacts, develop more excellent and stable experimental equipment, and be committed to realizing real-time monitoring and regulation of process parameters. In terms of materials, more research should be conducted on mixed powder materials with more stable physical and chemical properties, so as to reduce the vaporization induced by laser energy. Deeply understanding the vaporization phenomenon in the LPBF process requires more multi-scale modeling and simulation methods, from the micro-particle level to the whole process simulation of the macroscopic component size to predict and guide the optimization of printing parameters. More efforts should be made to develop the monitoring platform to realize the monitoring of the molten pool state while also being able to quantitatively measure the generation time of its defects and other effective data.

## Figures and Tables

**Figure 1 micromachines-15-00846-f001:**
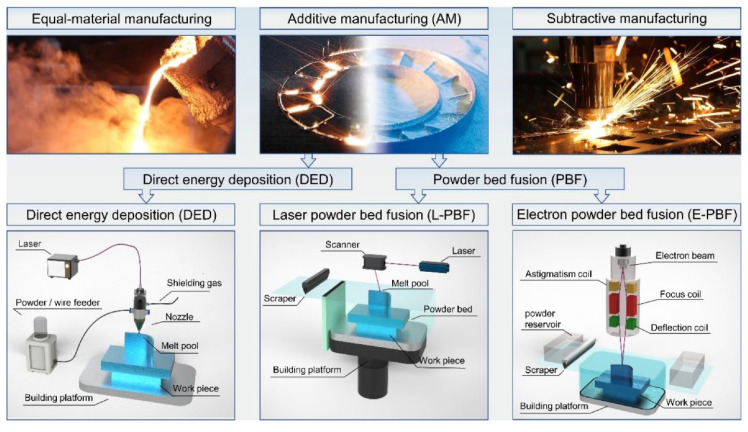
Classification of metal manufacturing processes: iso-material manufacturing, additive manufacturing, subtractive manufacturing [1].

**Figure 2 micromachines-15-00846-f002:**
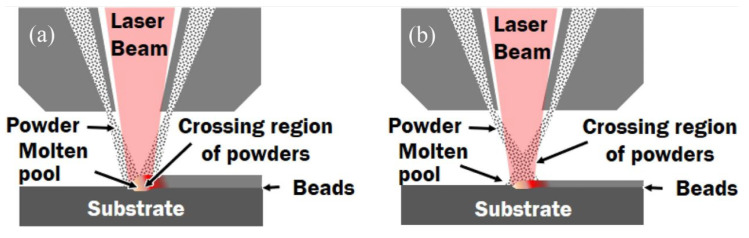
Powder feeding methodologies of LAM-DED processes: (**a**) co-axial feeding and (**b**) off-axis feeding [18].

**Figure 3 micromachines-15-00846-f003:**
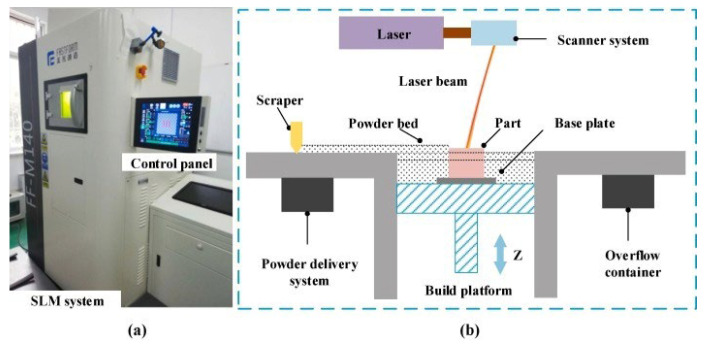
(**a**) LPBF device, (**b**) LPBF working diagram. Reprinted/adapted with permission from Ref. [21]. 2021, Elsevier.

**Figure 4 micromachines-15-00846-f004:**
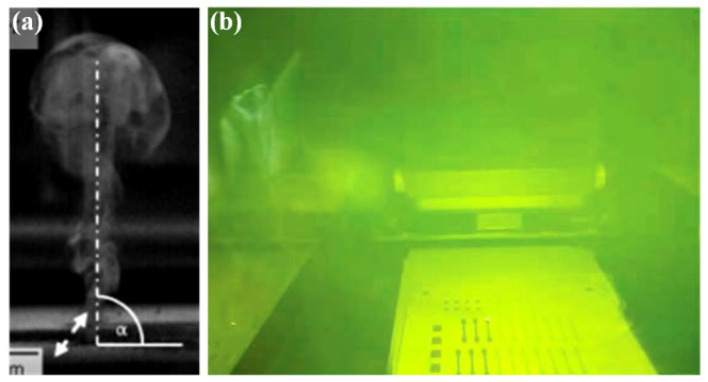
(**a**) Plume [38], (**b**) soot in molding bin.

**Figure 5 micromachines-15-00846-f005:**
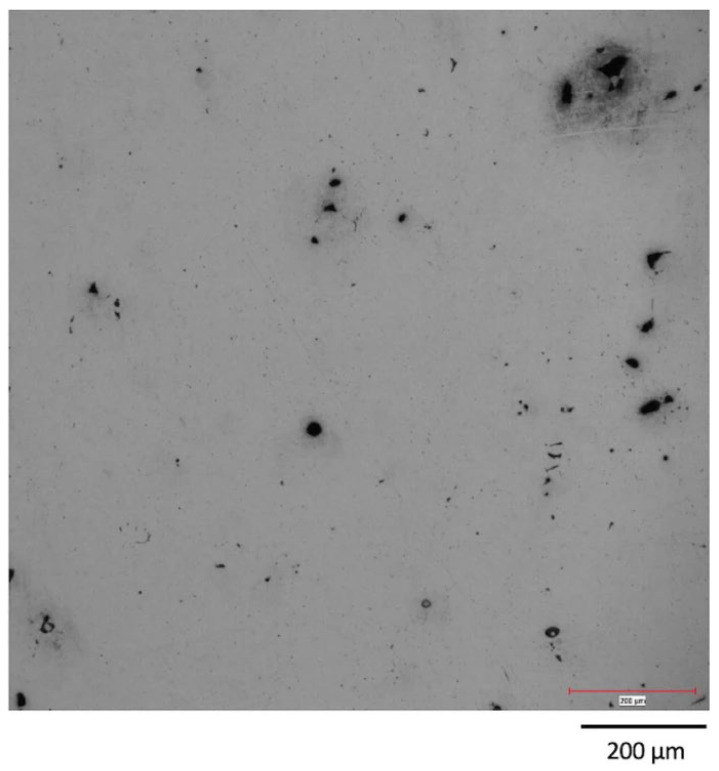
Microstructure of the pores [55].

**Figure 6 micromachines-15-00846-f006:**
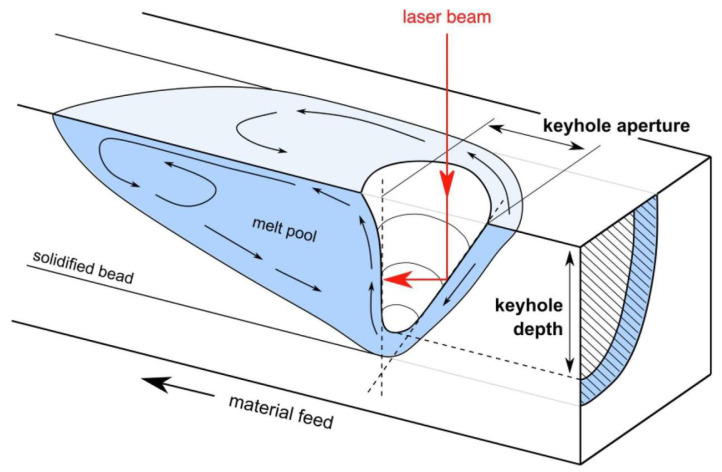
Keyhole [64]. Reprinted/adapted with permission from Ref. [64]. 2023, Elsevier.

**Figure 7 micromachines-15-00846-f007:**
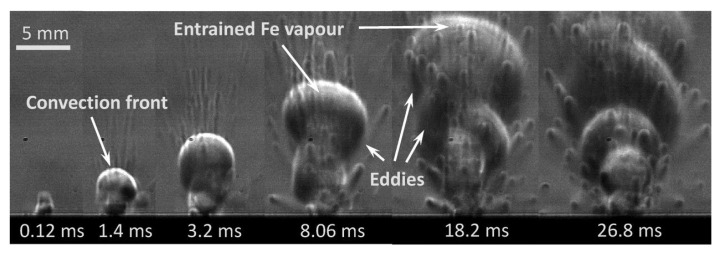
Plume forming process [80].

**Figure 8 micromachines-15-00846-f008:**
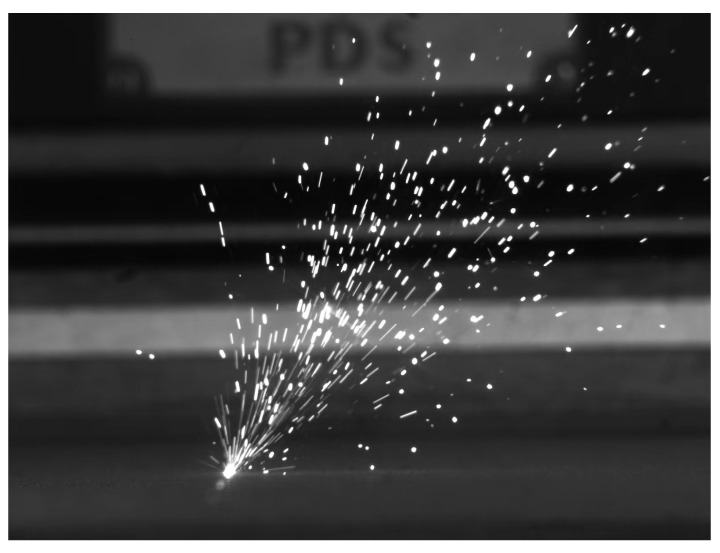
Schematic diagram of spattering.

**Figure 9 micromachines-15-00846-f009:**
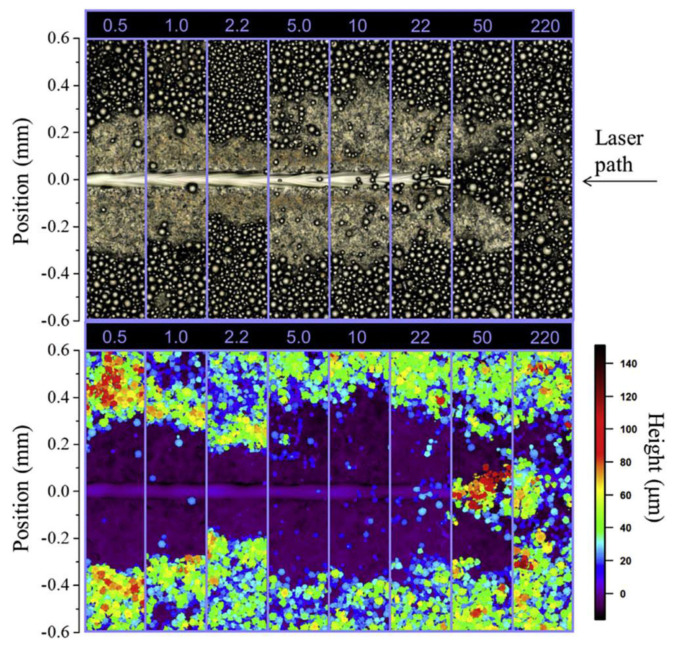
Area of erosion near the molten pool [90].

**Figure 10 micromachines-15-00846-f010:**
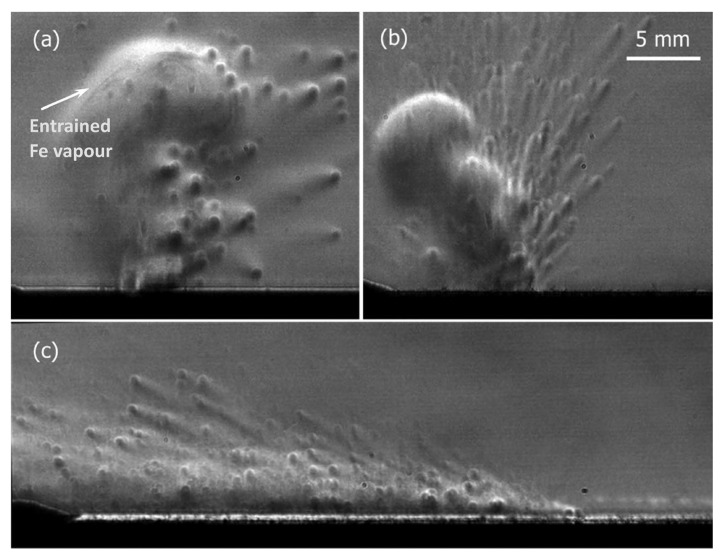
Schlieren images during left to right line scans at (**a**) 50 W and 0.1 m/s, (**b**) 100 W and 0.5 m/s, and (**c**) 200 W and 1 m/s. The characteristic refractive index gradients due to convection are visible in (**a**,**b**) but not present in (**c**) due to the backwards tilt of the laser plume. Entrained Fe vapour is visible behind the convection front in (**a**) [80].

**Figure 11 micromachines-15-00846-f011:**
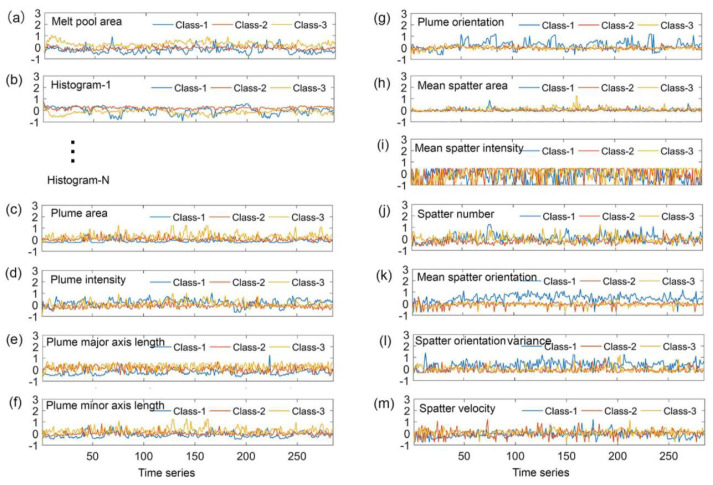
The extracted features from melt pool, plume and spatter. Reprinted/adapted with permission from Ref. [105]. 2018, Elsevier.

**Figure 12 micromachines-15-00846-f012:**
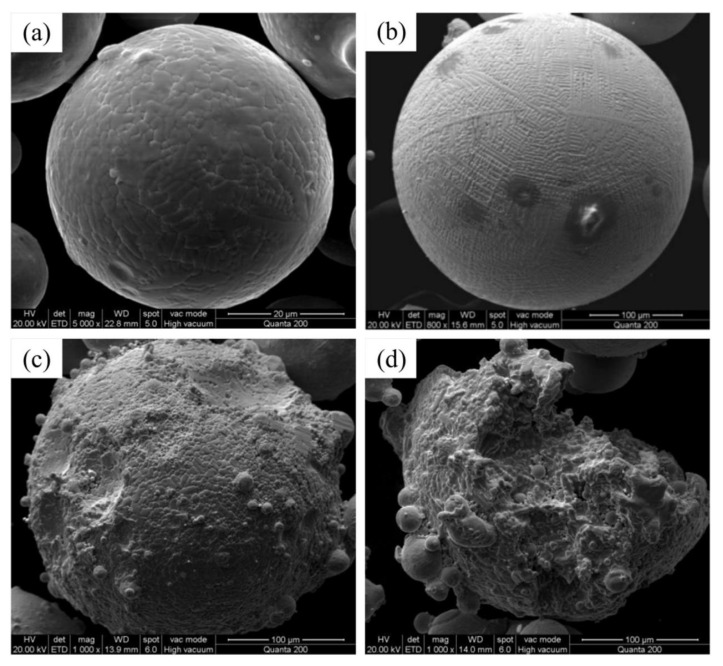
SEM images of initial powder and spattering particles. (**a**) Morphology of initial powder particles; (**b**) morphology of spherical spattering (Type I spattering); (**c**) coarse spherical morphology (Type II spattering); (**d**) irregular spattering (Type III spattering) [84].

**Figure 13 micromachines-15-00846-f013:**
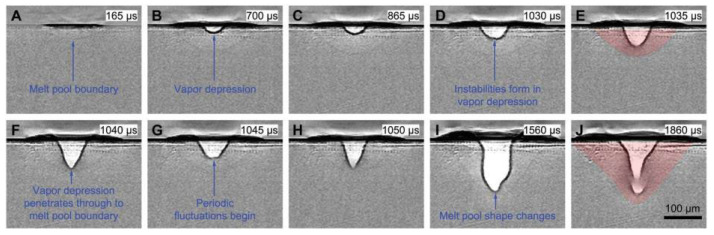
Evolutions of melt pool and vapor depression under stationary laser illumination. (**A**) Initial formation of a melt pool. (**B**) Formation of a small, stable vapor depression. (**C**) Steady growth of the vapor depression. (**D**) Instabilities form in the vapor depression. (**E**,**F**) Rapid change in the vapor depression shape. (**G**,**H**) Periodic fluctuation of the vapor depression. (**I**,**J**) Change of the melt pool shape from quasi-semicircular to bimodal with a bowl on top and a spike in the middle at the bottom. The sample is a Ti-6Al-4V bare plate. The laser spot size is 140 μm, and the laser power is 156 W. The images have been background-corrected by the image collected before the laser illumination. The shape of the melt pool is marked with a red shade in (**E**,**J**) [55].

**Figure 14 micromachines-15-00846-f014:**
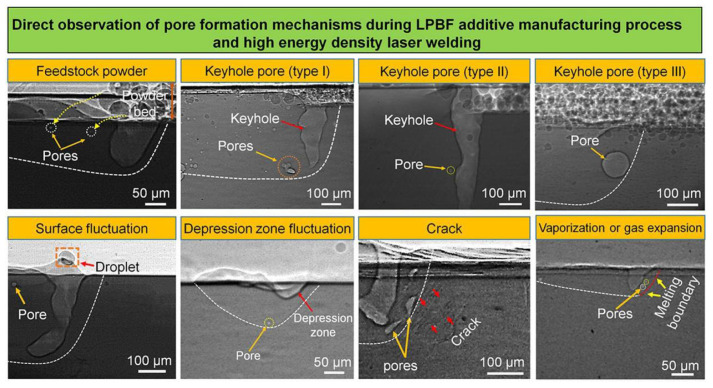
Different types of keyhole formation mechanisms. Reprinted/adapted with permission from Ref. [120]. 20201, Elsevier.

**Figure 15 micromachines-15-00846-f015:**
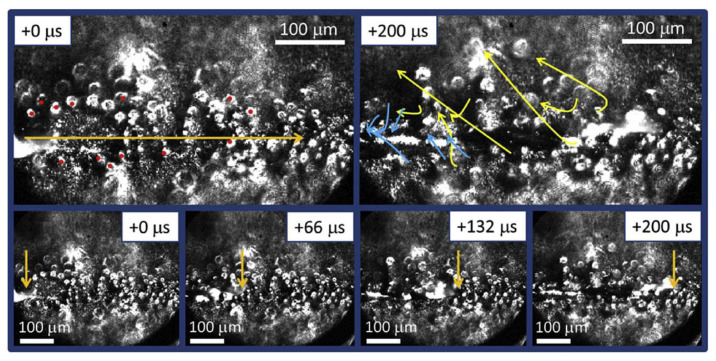
High-speed imaging of melt track progress and powder movement under the influence of the thermal steam Bernoulli effect [90].

**Figure 16 micromachines-15-00846-f016:**
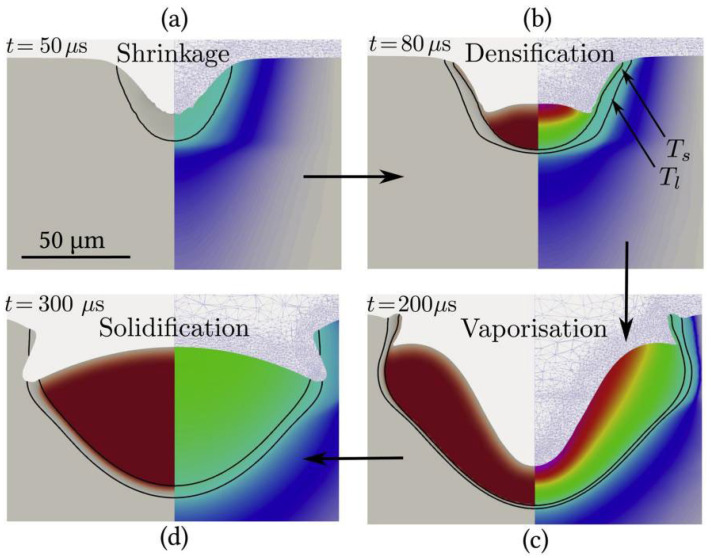
(**a**) Shrinkage of the powder; (**b**) densification of the powder into a molten pool; (**c**) vaporization phase with recoil pressure; (**d**) onset of solidification. Reprinted/adapted with permission from Ref. [127]. 2020, Elsevier.

**Figure 17 micromachines-15-00846-f017:**
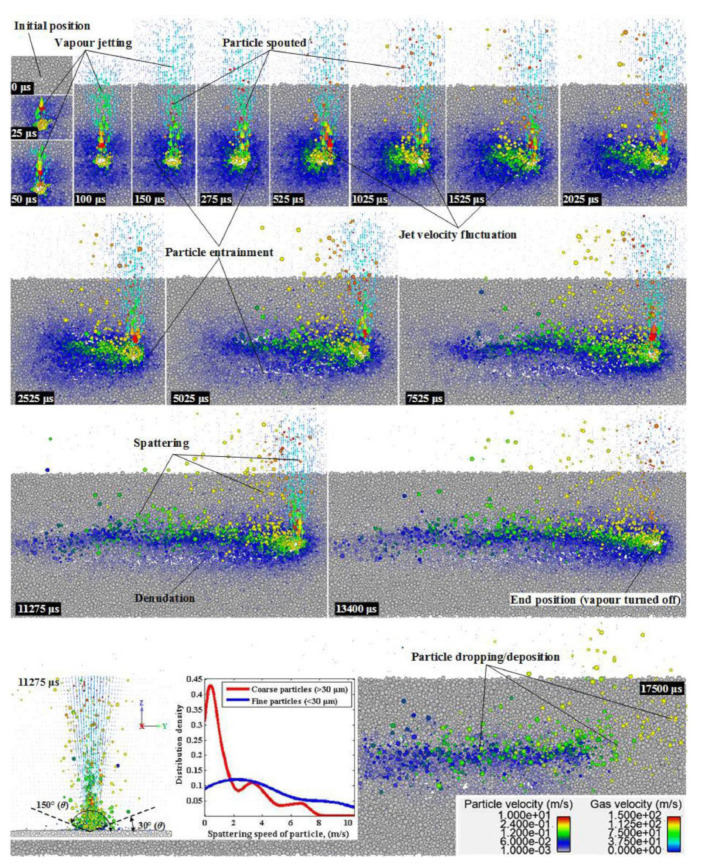
Numerical simulation of powder ejection. Reprinted/adapted with permission from Ref. [65]. 2020, Elsevier.

**Figure 18 micromachines-15-00846-f018:**
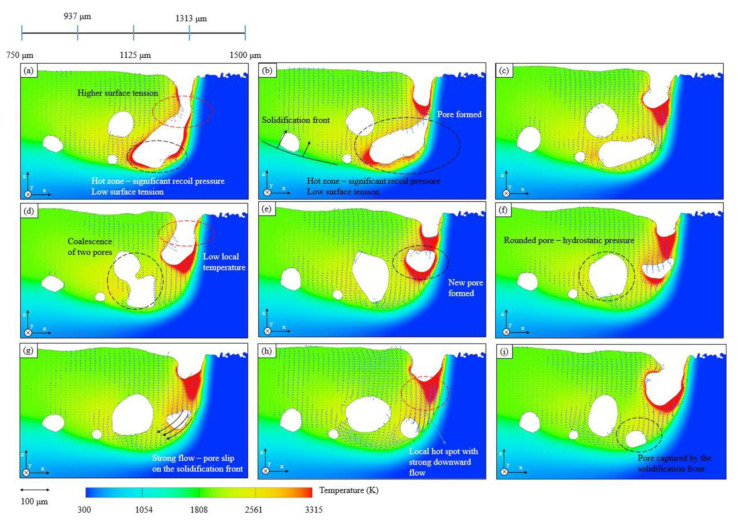
(**a**–**i**) Temperature cloud map of pore formation mechanism [131].

**Figure 19 micromachines-15-00846-f019:**
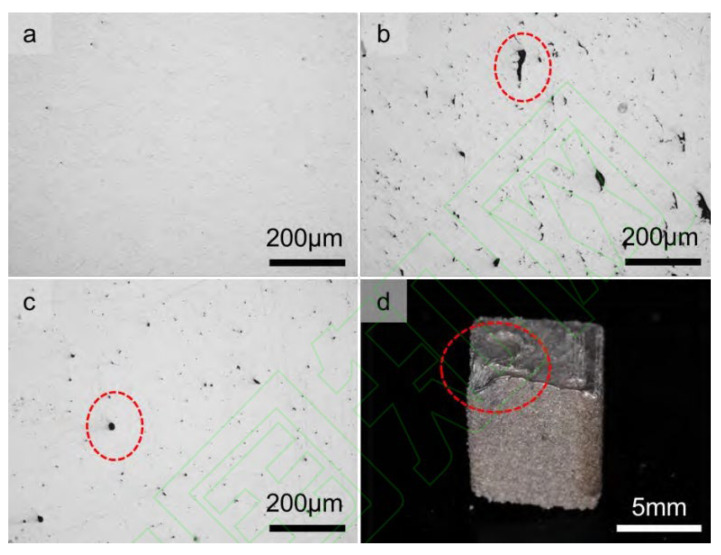
WE43 bulk section: (**a**) 80 W, 800 mm/s, (**b**) 40 W, 1000 mm/s, (**c**) 120 W, 600 mm/s; (**d**) part deformation caused by excessive heat input. Reprinted/adapted with permission from Ref. [134]. 2022, Chinese Journal of Lasers.

**Figure 20 micromachines-15-00846-f020:**
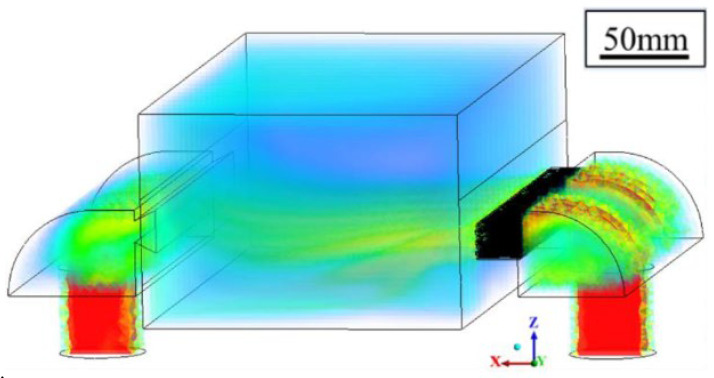
Working model of circulating air flow [136].

**Figure 21 micromachines-15-00846-f021:**
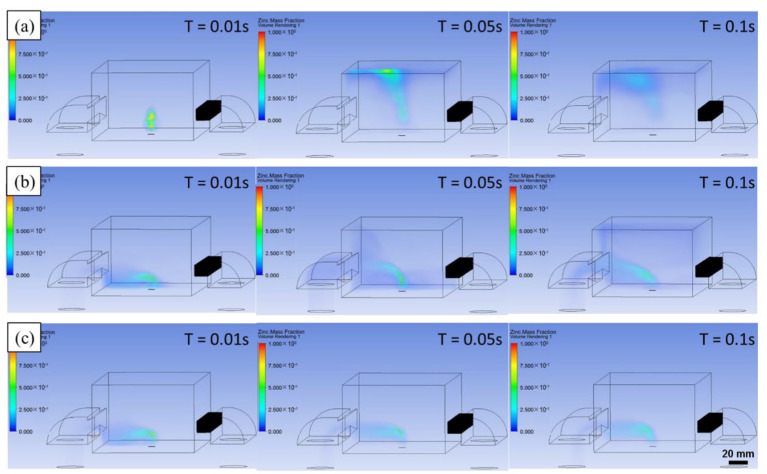
Effect of protective gas flow rate on evaporation smoke: (**a**) no protective gas flow rate; (**b**) insufficient protective gas flow rate; (**c**) sufficient protective gas flow rate [136].

**Figure 22 micromachines-15-00846-f022:**
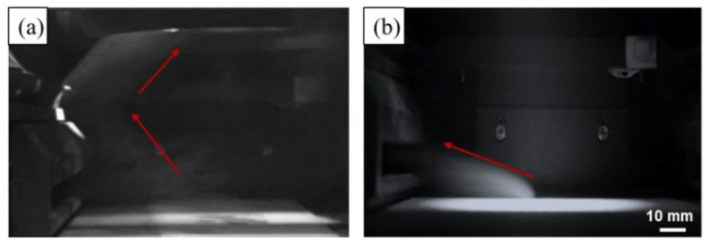
LPBF process captures Zn evaporation smoke: (**a**) insufficient dust removal air volume; (**b**) sufficient dust removal air volume [136].

**Figure 23 micromachines-15-00846-f023:**
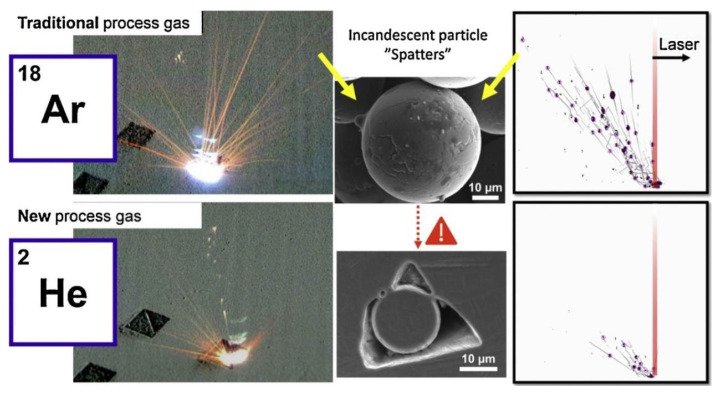
Spatter performance in different atmospheric conditions. Reprinted/adapted with permission from Ref. [139]. 2021, Elsevier.

**Figure 24 micromachines-15-00846-f024:**
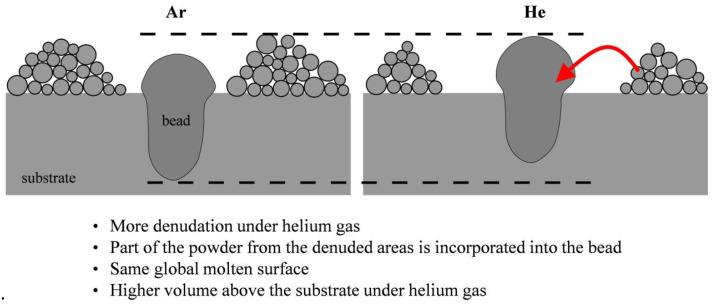
Bead morphology due to differences in denudation phenomenon in Ar and He atmosphere. Reprinted/adapted with permission from Ref. [140]. 2021, Elsevier.

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
