# Peer review of "A Review of the Vaporization Behavior of Some Metal Elements in the LPBF Process"

_micromachines, 2024, doi:10.3390/mi15070846_

Round 1
Reviewer 1 Report
Comments and Suggestions for Authors
1. The title is about vaporization behavior in the SLM process. but LPBF is often mentioned in the paper. Are the two processes exactly the same? Just different expression? Proposed unification.
2. Is Fig4 the cited paper? If so, cite the appropriate references.
3. The part about “Study of vaporization behavior“r is not summarized, just a simple list of references.
4. The format, size and font of the pictures in the paper are not uniform, so it is suggested to deal with the pictures uniformly.
5. Some of the references in this paper are too old, so it is recommended to cite new papers.
Author Response
Dear Reviewer:
We are extremely grateful that you took the time to review our manuscript and gave such detailed and valuable comments and suggestions.
In response to the question you raised, that is, the title involves the vaporization behavior in the SLM process, but LPBF is frequently mentioned in the text. Whether these two processes are completely the same, just with different expressions? We have conducted careful research and in-depth analysis, and now make the following responses and modifications: In the past, we usually abbreviated selective laser melting technology as SLM, but in recent years we generally use LPBF to refer to selective laser melting technology. In fact, they mean the same thing. In the new manuscript, in order to be more rigorous, we have changed the title to "A Review of the Vaporization Behavior of Some Metal Elements in the LPBF Process" (has been marked in red).
Regarding the question you mentioned, "Whether Figure 4 is a cited paper", it is indeed a literature that I cited. Due to my negligence, it was not marked. Currently, in the new manuscript, the referenced bibliography has been clearly indicated (has been marked in blue).
For the problem you mentioned, "The part of 'Research on Vaporization Behavior' has not been summarized, but just a simple list of references, and the pictures are not unified", we have attached great importance to it and carried out corresponding adjustments and improvements in the manuscript. We have deleted some pictures, and the newly added pictures have been marked in green. The adjusted parts of the main text are all in red font, and sections have been added at the end of each chapter to ensure the quality and scientific nature of the article.
Regarding the "old reference" problem you pointed out, in the new manuscript, we have deleted the old references and added several new references in recent years, which are 92, 95, 122, 123, 124, 127, 128, and 130 (has been marked in yellow).
Once again, we sincerely thank you for your professional guidance and hard work. We believe that through these improvements, the quality of the article will be further enhanced. If you have any other questions or need further information, please feel free to communicate with us at any time.
Sincerely, [Guanglei Shi]
[June 13, 2024]
Reviewer 2 Report
Comments and Suggestions for Authors
The authors intend to give a review on selective laser melting process. They have done a very good work on the search and retrieval of relevant publications on the field. However, they failed to transform this wealth of information into a useful review. The paper consists mainly of a pile of cited works without evidencing clearly the central points and their interrelation. There are too few paragraphs dividing the subtopics (e-g., the introduction is one paragraph spanning over one and a half pages).
The English is of low quality and repetitive. Many sentences are completely incomprehensible.
I cannot recommend the review for publication in the present form.
I suggest the authors to withdraw the paper and rethink the structure and means of presentations to obtain a useful review on this highly interesting field.
Comments on the Quality of English Language
The English is of low quality. The main issues are
- Incomprehensible sentences (e.g.; “The plume in the , we usually observe the plume from flow area, intensity of the plume, plume direction, plume long-axis length and plume short-axis length for the characterization of the plume.”)
- Repetitive use of phrases (e.g., “some – many – other scholars” appears at least twenty times
- Incomplete sentences
- Incorrect terms (e.g., “steam – smoke – vapor” are not equivalent)
- Unnecessary repetition of nouns (e.g.. “… from five aspects: plume area, plume intensity, plume direction, plume long-axis length, and plume short-axis length.”)
Author Response
Dear Reviewer:
We are filled with the most sincere gratitude and thank you for taking the precious time to review our manuscript and giving such meticulous and valuable opinions and suggestions.
In response to the issues you pointed out, that is, the failure to convert the abundant information into useful comments. This paper mainly consists of a pile of cited works put together, and there is no clear demonstration of the central point and its interrelationships, and there are too few paragraphs for subdividing subtopics. In this regard, after careful research and in-depth analysis, we have made the following responses and modifications: Firstly, for the paragraph issue, we have adjusted the main text and divided the overly long paragraphs into multiple parts to facilitate the readers' reading; Secondly, for the problem of not clearly demonstrating the central point and its interrelationships, we have added a lot of conclusive statements in the main text, and commented on the research methods and research contents of various scholars in this field, mainly analyzing their advantages and disadvantages, rather than simply piling up. All the relevant modified parts in the main text are marked in red font, and at the same time we have also deleted some pictures, and the newly added pictures have been marked in green.
Regarding the problem that "many sentences are difficult to understand" you put forward, it is indeed our oversight. We have used sentences that are more easily understood by the public to replace those rare words.
For the problem you mentioned that "the English quality is not good, there are many unnecessary repeated nouns and the important nouns are not unified", we have attached great importance to it and made corresponding adjustments and improvements in the manuscript. We have invited experts who are better at English to help us polish the overall wording and improve the coherence of sentences, and at the same time we have also deleted some overly colloquial nouns such as "some" and "many", and unified the important nouns in the article.
Once again, we sincerely thank you for your professional guidance and hard work. We firmly believe that through these improvements, the quality of the article has been further enhanced. If you have any other questions or need further information, you are welcome to communicate with us at any time.
Sincerely, [Guanglei Shi]
[June 13, 2024]
Reviewer 3 Report
Comments and Suggestions for Authors
The review article is well-written and has covered the vaporization behavior of some usual metal elements during selective laser melting process. I suggest it is acceptable after the following minor revisions.
1. The authors are encouraged to provide a section between abstract and introduction describing their search criteria for the review, inclusion and exclusion conditions and the justification for choosing these criteria.
2. The abstract is suggested to improve. The primary reviewed contents should be concretized in detail.
3. The discussion on vaporization behavior for some low-boiling materials such as Mg alloy during the LPBF process is vague, and clarity is needed in explaining to restrain vaporization behavior of Mg alloy and how it compares to other materials.
4. I would suggest adding a small section by the authors discussing the processing of additive manufacturing and challenges faced. Also, I suggest discussing the future directions. The prospects section may benefit from expansion to include discussions on future research directions, potential advancements, and the challenges that researchers need to address in the field.
5. The pictures are too messy, it must reorganize reasonably.
6. English need be carefully polished. For example, error in no comma before clause and first capital case of Equivalent in Line 25-26. Two full stop before However in Line 55.
Comments on the Quality of English LanguageEnglish need be carefully polished.
Author Response
Dear Reviewer:
We sincerely express our deepest gratitude thank you for devoting your precious time to reviewing our manuscript and offering such meticulous and highly valuable opinions and suggestions.
Regarding the issue you pointed out that "provide a section between the abstract and the introduction to describe the retrieval standards of its review, inclusion and exclusion conditions, and the reason for choosing this standard", after careful research and in-depth analysis by our team, the following responses and modifications have been made: In accordance with the requirements of the journal format, we cannot add the content of the retrieval standards for the cited literature between the abstract and the introduction. The literature we selected are all about the direction of additive manufacturing, and it is highly consistent with the theme of this article "Research on Metal Vaporization Phenomenon in LPBF Process". From these references, we can understand various generation mechanisms of the vaporization phenomenon and the research progress in current research internationally. We have integrated and summarized them, which can better help readers read.
For the suggestion you put forward that "improve the abstract and the main review content should be specified", it is indeed our oversight. We have re-revised the abstract and highlighted the core content described in this article. In the main text, we have also reconstructed the overall structure and added many conclusive contents. The focus is on conducting a review and evaluation of the research contents of current scholars. All the modified parts have been marked with red font.
Regarding the problem you mentioned that "the discussion about the vaporization behavior of some low boiling point materials such as magnesium alloys in the LPBF process is ambiguous and needs to be clearly explained to suppress the vaporization behavior of magnesium alloys and its comparison with other materials", we have attached great importance to it and made corresponding adjustments and improvements in the manuscript. We mentioned this problem in the subsection of the last chapter, that is, Chapter 4. For more volatile light metals (such as magnesium, aluminum), in fact, the suppression methods for the vaporization phenomenon of these materials are not much different from those of other materials, or in current research, no more effective suppression methods have been found.
For the series of problems such as "I suggest adding a subsection for the author to discuss the processing and challenges of additive manufacturing" you mentioned, we have summarized these problems in the last paragraph of the article, expounding the research dilemma in this field internationally at present and the general development direction in the future, etc.
Regarding the problem of "unified pictures", it is indeed our carelessness. We have re-adjusted the format of the pictures and deleted some useless pictures, while the newly added pictures have been marked with green font.
Finally, we invited experts who are better at English to help us polish the overall structure and language coherence of the article to avoid the recurrence of errors such as unsmooth sentences as last time.
Once again, we sincerely thank you for your professional guidance and hard work. We firmly believe that through these improvements, the quality of the article has been further enhanced. If you have any other questions or need further information, you are welcome to communicate with us at any time.
Sincerely, [Guanglei Shi]
Round 2
Reviewer 2 Report
Comments and Suggestions for Authors
I would like to congratulate the authors of this review on the tremendous work that they have done in the revision of the manuscript in the short time available. I have to admit that I doubted that such a profound reworking would be possible within the allotted time. The readability of the text has been dramatically improved with regard of the English and the organization of the content.
Minor issues with the English (mainly upper- and lower case) do not justify another revision but can be handled during the proofing process.
I recommend this new version of the review for publication.
Comments on the Quality of English Language
Minor issues with the English (mainly upper- and lower case) do not justify another revision but can be handled during the proofing process.
Reviewer 3 Report
Comments and Suggestions for Authors
Accept